# To Be Seen or Not to Be Seen: Latent Infection by Tobamoviruses

**DOI:** 10.3390/plants11162166

**Published:** 2022-08-21

**Authors:** Rabia Ilyas, Mareike J. Rohde, Katja R. Richert-Pöggeler, Heiko Ziebell

**Affiliations:** Institute for Epidemiology and Pathogen Diagnostics, Julius Kuehn Institute, Federal Research Institute for Cultivated Plants, Messeweg 11-12, 38104 Braunschweig, Germany

**Keywords:** latent tobamoviruses, asymptomatic infection, host–virus interaction

## Abstract

Tobamoviruses are among the most well-studied plant viruses and yet there is still a lot to uncover about them. On one side of the spectrum, there are damage-causing members of this genus: such as the tobacco mosaic virus (TMV), tomato brown rugose fruit virus (ToBRFV) and cucumber green mottle mosaic virus (CGMMV), on the other side, there are members which cause latent infection in host plants. New technologies, such as high-throughput sequencing (HTS), have enabled us to discover viruses from asymptomatic plants, viruses in mixed infections where the disease etiology cannot be attributed to a single entity and more and more researchers a looking at non-crop plants to identify alternative virus reservoirs, leading to new virus discoveries. However, the diversity of these interactions in the virosphere and the involvement of multiple viruses in a single host is still relatively unclear. For such host–virus interactions in wild plants, symptoms are not always linked with the virus titer. In this review, we refer to latent infection as asymptomatic infection where plants do not suffer despite systemic infection. Molecular mechanisms related to latent behavior of tobamoviruses are unknown. We will review different studies which support different theories behind latency.

## 1. Introduction

The genus *Tobamovirus* comprises positive-sense, single-stranded RNA (+ssRNA) viruses belonging to the family *Virgaviridae* and contains 37 recognized and several tentative species [1]. These viruses can infect a wide range of host plants, the type member tobacco mosaic virus (TMV) alone is known to infect approximately nine plant families and at least 125 individual plant species including tobacco, tomatoes, cucumbers and orchids [2].

### 1.1. Disease Symptoms

Tobamovirus disease symptoms can vary depending on the host plant, virus species and environmental conditions. Leaves of infected plants appear to be deformed, mottled or display a mosaic pattern [2]. Sometimes, in case of severe infection, systemic necrosis and defoliation occurs depending on biological (age of plant, cultivar, virus strain, etc.) and environmental factors (temperature, light intensity, etc.) [3]. TMV and tomato mosaic virus (ToMV) are the most extensively studied species and they commonly cause chlorosis, mosaic patterns and leaf distortion on susceptible hosts (Figure 1). Severe infections can also lead to systemic necrosis and defoliation. Pepper mild mottle virus (PMMoV) and tobacco mild green mosaic virus (TMGMV) infect vegetables in the family *Solanaceae*. PMMoV usually causes milder symptoms on leaves but is more severe on fruits causing a reduction in size and number, uneven ripening, corky or necrotic rings and internal necrosis. Tomato brown rugose fruit virus (ToBRFV) was first described by Salem et al. [4] and, since then, has become a threat in tomato growing regions of the world [5]. The symptoms appear both on leaves and on fruits causing deformation, yellowing and necrosis [6]. The leaves show interveinal yellowing and mosaic patterns while the fruits show marbling and discoloration, which reduces the market quality of the fruits [4,5,7].

Apart from the agriculturally important crop plants, several new tobamoviruses have been detected infecting horticultural crops. Two tobamoviruses have been identified from hibiscus (Hibiscus latent Singapore virus (HLSV) and hibiscus latent fort pierce virus (HLFPV) [7,8]) Similarly, two tobamoviruses have been reported from wax flower plants belonging to *Hoya* spp. showing chlorotic ring spots and irregular leaf chlorosis. [9,10]. 

### 1.2. Genome Organization

Tobamoviruses are one of the most studied viruses due to their availability and ease of use in biotechnological studies [11]. A single virion of TMV is 18 nm wide and 300–310 nm long. The encapsidation of subgenomic RNA (sgRNA) results in shorter virions of 32–34 nm length, but these are a minor component of the total virion population [1]. The rod-shaped virions have an inner core of 4 nm diameter, which contains a +ssRNA molecule (Figure 2). The outer core is made up of 2100 subunits of coat protein. These subunits are arranged in the form of a right-handed helix around the RNA molecule [12]. Species demarcation criteria for tobamoviruses are based on nucleotide sequence similarity of the total genome. Less than ten percent difference in nucleotide sequence is defined to identify strains of a species [1].

The extreme stability of tobamoviruses and their presence in the environment has been attributed to their structure [13]. TMV particles were found to be stable after heating for 30 min at 60 °C, at pH ranges between 2–10 and in organic solvents of up to 80% by volume. Particles lose their protein subunits from the 5′ terminal of RNA in alkaline solutions of pH 9 and above. 

Purified TMV can be infectious even after 50 years of storage at 4 °C [13]. TMV has been persistently found in the environment including clouds, water, glacial ice and soil [14]. The persistence of tobamoviruses in water may contribute to their dissemination in the environment, as well as agricultural systems. Tobamoviruses have been detected in natural water bodies near and far from agricultural production in Europe and North-America [15]. 

The +ssRNA genome is approximately 6300 nt to 6800 nt in length. Figure 3 shows a schematic representation of the genome organization of TMV. Two overlapping ORFs begin at the 5′ proximal start codon. Termination at the first in-frame stop codon produces a 125–130 kDa protein. A 180–190 kDa protein is produced by read-through of its leaky termination codon approximately 5–10% of the time. These two proteins are required for replication and are produced from the genomic RNA. In addition, three more proteins are translated from two co-terminal sgRNAs of approximately 1.6 kb and 0.7 kb in length [16]. 

The next ORF encodes the 28–34 kDa movement protein, which has RNA-binding activity and is required for cell-to-cell movement [17,18]. A positively charged protein was also found to be expressed from the MP-associated sgRNA [19]. This 4–8 kDa protein has been shown to increase the virulence in *N. benthamiana* and may be associated with cellular movement [20]. The 3′ proximal ORF encodes a 17–18 kDa coat protein (CP). A sgRNA containing an ORF for a 54kDa protein that encompasses the read-through domain of the 180–190 kDa ORF has been isolated from infected tissue, although no protein has been detected [12,21]. 

The 5′-UTR region of TMV starts with an m^7^G-capped G residue, which is followed by a ~70 nt G-deficient sequence, called an omega sequence (Ω), that contains multiple CAA repeats. The sg RNAs also contain these motifs [18]. Several tobamoviruses contain such CAA repeats in 5′-UTR [22]. The 3′-UTR in TMV RNA lacks a 3′poly-(A) tail and contains sequences that can be turned into a series of pseudoknot structures, followed by a tRNA-like terminus (Figure 4). This addition of 204 nt in 3′-UTR stabilizes the mRNA and enhances translation [22]. These UTR regions play multiple roles in translation and replication of tobamoviruses. Recently, 2 tobamoviruses from hibiscus have been reported to contain an internal 3′-poly(A) tail region [8,23,24,25] raising questions about functions and diversity of UTRs in tobamoviruses. 

### 1.3. Subgroups

On the basis of genome organization, tobamoviruses have been divided into three subgroups [26,27,28]. These subgroups are based on the location of origin of assembly (OA) and overlapping open reading fames ORFs (Table 1). Three cactus-infecting members of the genus, however, are only distantly related to the other members and, therefore, come as an outgroup in the phylogenetic analysis (Figure 5). Min et al. [29] suggest a new subgroup for these viruses. 

### 1.4. Transmission

The spread of tobamoviruses is mainly driven by mechanical transmission. Seed transmission is also possible. Tobamoviruses do not infect the embryo but can be transmitted to the seedling from the surrounding tissue during germination. Tobamoviruses can be found in the seed coat and sometimes in the endosperm [30]. Seed transmission from outside of the embryo requires a high stability of the viral particles and does not occur at a high rate [31]. However, seed transmission is an important factor in the dissemination of tobamoviruses such as ToBRFV.

Other sources of primary inoculum include the juice of infected fruits, as well as plants grown in infected soil or grafted with infected material [32]. Bumblebees, which are commonly used for pollination in tomato crops, have also been shown to carry ToBRFV particles and hives from affected greenhouses can introduce the virus into new facilities [32]. 

### 1.5. Symptoms—To Be Seen or Not to Be Seen: The Concept of Viral Latency and Asymptomatic Infection

As a part of the disease triangle, host, virus, and environment equally play a role in disease establishment. Therefore, the viral symptoms are dependent on the right host and a suitable environment. Over the last few decades, there have been various meanings and explanations for asymptomatic infection and virus latency. This terminology is loosely used in the context of a host–virus interaction that does not give rise to visible disease symptoms as shown in Table 2. Roger Hull explains latency as a consequence of tolerance: where virus is able to replicate and move systemically in the host with little or no impact on the plant’s overall health [33]. On the other side, Takahashi et. al. defines latency as a phenomenon when the virus is in a dormant state and does not replicate in the host [34]. They discuss that the asymptomatic infection can either result from tolerance or persistence. In this review, we refer to latent infections as asymptomatic infections where plants do not develop symptoms despite systemic infection.

### 1.6. Mechanisms That May Explain Latent/Asymptomatic Infection

The existence of several definitions of latent infection gives rise to even more explanations of mechanisms that may cause this phenomenon. According to Roger Hull [33], there can be six possible reasons for an asymptomatic infection: (1)Infection with a very mild virus strain,(2)A tolerant host plant,(3)Leaves that escape infection because of their age and position on the plant,(4)‘Recovery’ from the disease symptoms in newly formed leaves,(5)Dark green areas in a mosaic pattern,(6)Plants that are infected with cryptic viruses.

Viral populations encompass different genetic variants, and their distribution is shaped by selection and genetic drift. In addition to virus replication and maintenance, properties of the host plant play an important role for the selection of viruses. This includes the adaptation to plant resistance, plant defense mechanisms and also the relationship between virus replication and plant fitness. An interesting aspect of viral genome organization are the multifunctional proteins and overlapping open reading frames. This results in a trade-off regarding selection for different functionalities. Although RNA viruses can show high mutation rates, their proteins are not more variable than those of DNA-viruses [35,36]. An independent assessment of mutation rates without the influence of selection is difficult. A mutation rate of 0.02–0.05, with multiple mutations in 35% of cases and 69% of indels was found in the mutational spectrum [37]. The recombination of RNA viruses is also an important source of genetic variation and has been shown to occur in tobamoviruses in the case of TMV and ToMV [38].

Genetic variation can lead to differences of symptom severity in plants infected with different strains. For example, it has long been known that attenuated strains of TMV exist, which cause little or no symptoms in tobacco plants and may even interfere with the development of symptoms in a mixed infection with a more severe strain [39,40]. In some cases, changes in symptom severity could be associated with specific mutations. For example, recovery mutants of the mild strain TMV-K were used to pinpoint nonsense mutations in the replicase and movement protein [41], similar mutants are known for other tobamoviruses [42,43]. On the other hand, mutations can also lead to increased virus reproduction and/or symptom development. An example is an increase of TMV replication due to disruptions in its the secondary structure [44]. Since the expression of sub-genomic viral RNA is regulated by sub-genomic promoter and enhancer elements, disruption of the relevant motifs also leads to changes in protein expression [45]. These regions might also be targets of alleviating mutations. 

Most studies on tobamoviruses focus on virus multiplication as the criteria for trade-off between host–virus interactions. However, when it comes to host range determination, there are two important components of viral fitness: particle stability and infectivity [46]. As shown by a series of studies, maintaining the three-dimensional structure of TMV CP was essential for N-gene resistance elicitation [47]. Therefore, resistance breaking is associated with altered particle stability. A trade-off between increased viral reproduction and extended survival can lead to slower rates of replication for enhanced particle stability [46]. Fraile et al. explain infectivity as the relationship between inoculum dose and infection success [46]. Mutational analysis experiments from Culver et al. showed increased particle stability correlated with decrease in infectivity and less efficient viral translation [48]. 

Another reason that might explain latency can be the balance between host defense and viral counter defense. Tobamoviruses have co-evolved with their hosts [49] and have developed sophisticated mechanisms to counteract the host defense systems. It might be possible that these finely tuned interactions allow some tobamoviruses to persist in hosts without causing much damage. Naturally occurring latent tobamoviruses have not been studied extensively. There are studies on TMV that support the idea that plant viruses can suppress host defense signaling by promoting the degradation of ATF2, a plant NAC transcription factor, which regulates the expression of PTI-responsive genes [50].

The beneficial interaction between plant viruses and hosts have been discussed in more detail in previous studies [51,52]. Experimental evidence supports the beneficial impact of accommodating long-term virus infection, especially in natural environments [52]. TMV infection has shown to improve the plant resilience in sub-optimal environmental conditions, for example, tolerance to drought [53]. Virus-induced drought tolerance is associated with global reprogramming of plant gene expression, changes in hormone signaling and increased accumulation of metabolites and antioxidants [20]. Interestingly, recent studies suggested that the benefits of increased drought resistance can be offset by increased virus virulence [20]. Maintaining persistent virus infection can also improve the plant resistance to biotic stress including non-vector herbivory insects, other viruses or unrelated pathogens [52,54].

### 1.7. Potential Usefulness of Latency

Many wild plants naturally harbor more than one virus. Synergistic or antagonistic interactions between two or more viruses in a host can alter symptom development. A synergistic interaction between two viruses can improve vial replicability and fitness inside the host. A detailed discussion about antagonism and synergism between plant viruses is available in previous reviews [55,56]. Antagonistic interactions can result in superinfection of one virus hindering replication and accumulation of a second virus. Such an interaction can result in mild or no symptoms on the host plant. An example for such a case is the cross-protection provided by HLSV against TMV in *N. benthamiana* [57]. HTS of these plants revealed that TMV-vsiRNA possessed high sequence complementarity to a host gene which encodes a C2-domain abscisic acid (ABA)-related (CAR) 7-lke protein. CAR proteins play a crucial role in the ABA signaling pathway [44]. If an acute virus infection can have a more complex host–virus relationship than a disease damaging its host, it is perhaps easier to imagine that an asymptomatic infection can turn out to be a mutualistic relationship with the host. More research is needed to fully understand these multifaceted interactions. 

On the other side, viruses that cause mild or no symptoms have been widely studied for their potential use in mild strain cross-protection. It is a well-known phenomenon that plants are protected from severe virus strains due to the pre-infection with a mild strain of the same virus [58,59]. The term mild strain can be used for asymptomatic, mutated or attenuated virus strains [60]. An example for tobamoviruses is the cucumber green mottle mosaic virus (CGMMV) where the VIROG-43Ms strain of CGMMV provided protection in cucumber plants against the MC-1 and MC-2 strains of the same virus [58]. A somewhat similar concept applies to the cross-protection conferred by the latent infection-causing viruses against the acutely-infecting viruses of the same genus. For example, Hibiscus latent Fort-Pierce virus (HLFPV) reduced the intensity of symptoms when it was inoculated on *N. benthamiana* plants against the U1 strain of TMV [57]. 

### 1.8. Potential Danger of Latency

Despite the above-mentioned benefits of asymptomatically present plant viruses, their potential to spread to a different host and cause diseases in a new environment cannot be ignored. Viral disease symptoms are often host- and environment-dependent. Therefore, when inoculated on a susceptible host in a new environment, an asymptomatic virus can potentially induce severe symptoms [60]. The threat is even bigger in the case of vegitatively propagated crops. Latent infections can go undetected and spread across the globe. One such example is the Hoya tobamovirus-2, which was detected in *hoya* plants using a combined EM- HTS approach [9]. The plants showed necrotic lesions and ringspots due to the mixed infection with tomato spotted wilt virus (TSWV) belonging to *Orthotospovirus* genus. HoTV-2, however, did not cause symptoms when *N. benthamiana* plants were mechanically inoculated with the virus (case study Ilyas 2021). 

### 1.9. Examples of Latent Infections Regarding Tobamoviruses

While most common members of the tobamovirus genus cause severe disease symptoms in host plants, there are other members, which do not cause obvious symptoms in wild host plants. Examples of latent tobamoviruses include Hibiscus latent Fort-Pierce virus [61], Hibiscus latent Singapore virus, tobacco latent virus [62], Brugmansia latent virus and Hoya tobamovirus-2 [9]. 

#### 1.9.1. Tobacco Latent Virus (TLV)

Tobacco latent virus belongs to the subgroup I (solanaceous-infecting) tobamoviruses. The virus was found in Nigeria, in field-grown tobacco from a mixed infection with several other viruses [62]. It remained symptomless in several cultivars of *N.tabacum*, which were systemically infected [62], while its closest relative, tobacco mild green mosaic virus (TMGMV), causes clear disease symptoms and yield losses in pepper crops worldwide [63]. 

#### 1.9.2. Brugmansia Latent Virus (BLV)

*Brugmansia* is an ornamental plant belonging to the *Solanaceae* family. During an examination of the *Brugmanisa* plant collection in the Royal Botanical Garden Kew, for Columbian datura virus (CDV), Brugmansia latent virus was detected in plants with no obvious viral symptoms [64]. The virus was detected in multiple asymptomatic hosts when the plants were tested with reverse transcription polymerase chain reaction (RT-PCR) using generic tobamo primers. The virus belongs to subgroup I of tobamoviruses and is closely related to bell pepper mottle virus (BPMV) and tomato mosaic virus (ToMV) which induce mottling and mosaic symptoms on host plants. 

#### 1.9.3. Latent Infection Causing Viruses from Hibiscus

Hibiscus is used as an ornamental plant mostly in tropical and subtropical climates for hedges and flowers. Plants from *Malvaceae* family have not been known as common hosts for tobamoviruses until the detection of two latent viruses from hibiscus plants [61]. Hibiscus latent Singapore virus (HLSV) and Hibiscus latent Fort Pierce virus (HLFPV) were both reported to be symptomless on several experimental and natural hosts [8,65]. HSFPV was also found in mixed infections with other hibiscus infecting viruses, which resulted in chlorotic ring spots. 

These two viruses are unique among the other members of this genus because of their internal poly-(A)-tract (IPAT) in the 3′-UTR region [8,23]. This IPAT has variable lengths between 77 to 96 nt. The IPAT functions similarly to the upstream pseudoknot domain (UPD) of TMV 3′-UTR [24]. The IPAT of minimal 24 nt was found essential for HLSV RNA replication and CP expression whereas TMV UPD is not essential for CP expression and systemic movement in *N. benthamiana* [24]. The presence of IPAT in both tobamoviruses from *Hibiscus* suggest that this is a common feature among the malvaceous-infecting tobamoviruses. Niu et al. [24] speculated that the ancestors of tobamoviruses originally possessed UPDs. During evolution, due to host switching, the ancient tobamovirus may have acquired an IPAT through intermittent template switching by host mRNAs during virus replication. This way, the newly evolved virus could replicate well in its new host and became the dominant species in hibiscus. 

Viruses that possess an IPAT seem to cause milder or no symptoms in their hosts [24]. This even extends to the mutants of acutely infecting viruses. An IPAT containing TMV hybrid (TMV 43A) showed very mild symptoms in *N. benthamiana*. [24]. It is possible that the IPAT can reduce symptom development by downregulating viral gene expression. 

The phylogeny (in Figure 3) of the above-mentioned latent viruses shows no uniform pattern that could explain their differential behavior as compared to other tobamoviruses which cause significant economic losses in crop plants. The presence of latent viruses seems to be common among all tobamovirus subgroups (Figure 6), but they were often undetected due to bias in detection assays, as well as a preference to study viruses causing disease in agricultural crops. In the last few years, many viruses causing latent infections have been detected due to HTS [9,66,67,68]. 

**Table 2 plants-11-02166-t002:** Terminology used in the context of asymptomatic infection and their explanation in the case of tobamoviruses.

Terminology	Explanation	Reference	Outcome	Example in Tobamoviruses
Latency	The virus can replicate and move systemically but does not cause disease	[33]	No visible symptoms	Hibiscus latent Fort Pierce virus [10] Hibiscus latent Singapore virus [8] Tobacco latent virus [62] Brugmansia latent virus [64]
Tolerance	The virus is able to replicate but host development is not much impacted despite high virus titer	[34]	Mild symptoms	Unknown
Persistence	The virus is able to replicate in host, but the titer remains low	[34]	No symptoms	Unknown
Endogenous viruses	Virus is integrated into the host genome. Some can be activated under certain conditions.	[69]	Usually, no symptoms	Unknown
Immunity (non-host)	The virus is unable to replicate in host cells	[33]	No symptoms	Tobacco mild green mosaic virus in tomato [70]
Resistance (host) (Hypersensitive response)	The virus can replicate in initially infected cells. Viral movement is limited to the surrounding cells. Visible necrotic local lesions, plants are field resistant.	[33]	Small necrotic lesions	N-gene resistance in tobacco plants against TMV [71]
Susceptibility	The virus can replicate and cause disease	[33]	Visible disease symptoms	TMV infection in petunia [72]

## 2. Concluding Remarks

Although latency has been observed for many tobamoviruses, the mechanisms underlying this phenomenon are still elusive. While there are multiple definitions of latent infection in the literature, there are even more explanations that use this umbrella term as mentioned in Table 2. With respect to our understanding of tobamoviruses, there are still many knowledge gaps regarding latent infection, as well as viruses infecting ornamental plants. While it is important to study new and emerging tobamoviruses that are threatening horticulturally important crop plants, it is equally important to investigate alternative host plants and to understand tobamoviruses that cause a latent infection that can go unchecked and potentially cause diseases on other hosts. 

### Future Aspects

i.There is a need to combine HTS with biological, serological, electron microscopy and molecular detection and characterization methods to reliably detect the asymptomatic viruses.ii.Further studies are important to understand why some tobamoviruses cause latent infection and if that latency is host dependent.iii.For crop protection, it is important to analyze if latent viruses from ornamentals can infect crop plants in different climatic and geographical conditions.iv.Latent tobamoviruses can be used as a model for studying host–virus interaction and virus–virus interaction in a cross-protection scenario.

## Figures and Tables

**Figure 1 plants-11-02166-f001:**
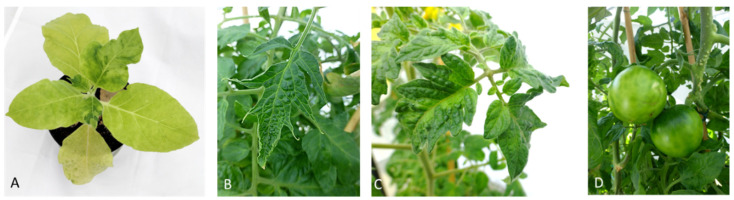
(**A**) TMV symptoms on *Nicotiana tabacum* cv. ‘Samsum nn‘, showing leaf distortion, mosaic pattern and chlorosis; (**B**–**D**) ToBRFV symptoms on tomatoes showing mosaic pattern, leaf narrowing and rugose spots on fruits.

**Figure 2 plants-11-02166-f002:**
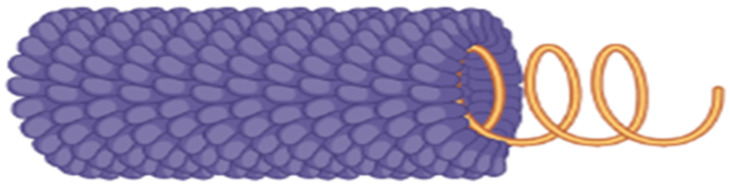
Structural diagram of a TMV particle. Protein subunits are shown in purple while the ssRNA is shown in orange. Source: Image created in Biorender.

**Figure 3 plants-11-02166-f003:**
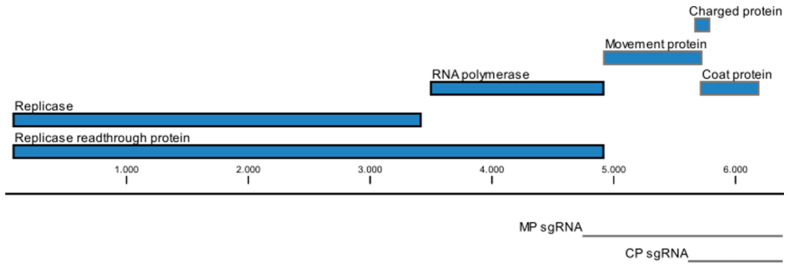
Schematic diagram of genome organization of TMV: ORFs encoding viral proteins are shown in blue. The 126 kDa replicase protein and the 183 kDa replicase read-through protein are translated from the genomic RNA (black), while the remaining proteins are expressed from separate subgenomic RNAs (gray). Image was modified from the NCBI Reference Sequence (NC_001367.1) [16] using CLC main workbench 21 and Inkscape 1.1.

**Figure 4 plants-11-02166-f004:**
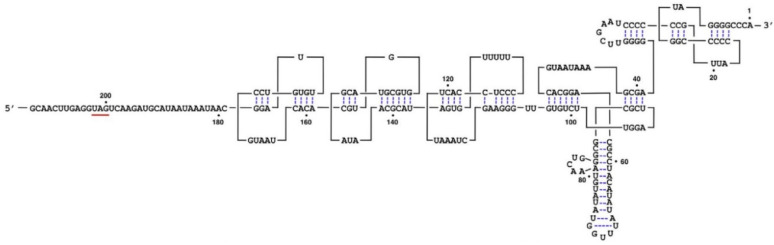
Schematic diagram of 3′-UTR of TMV RNA. Translational stop codon is underlined and nt numbers from the 3′ region are indicated. Image source [22].

**Figure 5 plants-11-02166-f005:**
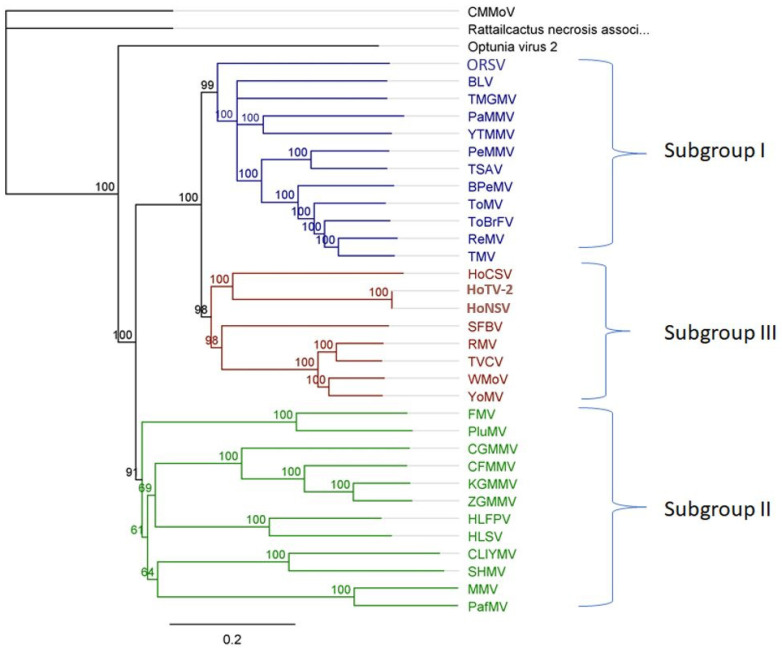
Phylogenetic analysis of tobamoviruses based on whole genome organization. Different colors represent subgroups within the genus. Unrooted neighbor-joining trees were constructed using Geneious (version 8.9.1). Branch lengths indicate the number of nucleotide differences per site and the numbers at each node indicate the bootstrap values. CMMoV: cactus mild mottle virus (accession: NC_011803.1), rattail cactus necrosis associated virus (Accession: NC_016442.1), Optunia virus 2 (accession: MF434821.2), ORSV: Odontoglossum ringspot virus (accession: E04305.1), BLV: Brugmansia latent virus (accession: MK012556), TMGMV: tobacco mild green mosaic virus (accession: NC_001556.1), PaMMV: paprika mild mottle virus (accession: NC_004106.1), YTMMV: yellow tailflower mild mottle virus (accession: NC_022801.1), PMMoV: pepper mild mottle virus (accession: NC_003630), TSAV: tropical soda apple mosaic virus (accession: NC_030229), BPMV: bell pepper mottle virus (accession: NC_009642), ToMV: tomato mosaic virus (accession: NC_002692), ToBrFV: tomato brown rugose fruit virus (accession: NC_028478), ReMV: Rehmannia mosaic virus (accession: NC_009041), TMV: tobacco mosaic virus (accession: NC_001367), HCSV: Hoya chlorotic spot virus (accession: KX434725), HoTV-2: Hoya tobamovirus-2 (accession: MT750216.1), HoNSV: Hoya necrotic spot virus (accession: MN961200.1), SFBV: Streptocarpus flower break virus (accession: NC_008365), RMV: ribgrass mosaic virus (accession: NC_002792), TVCV: turnip vein clearing virus (accession: NC_001873), WMoV: wasabi mottle virus (accession: NC_03355), YoMV: youcai mosaic virus (accession: NC_004422), FrMV: frangipani mosaic virus (accession: NC_014546), PluMV: Plumeria mosaic virus (NC_026816), CGMMV: cucumber greem mottle mosaic virus (accession: NC_001801), CFMMV: cucumber fruit mottle mosaic virus (accession: NC_002633), KGMMV: kyuri green mottle mosaic virus (accession: NC_003610), ZGMMV: zucchini green mottle mosaic virus (accession: NC_003878), HLFPV: Hibiscus latent Fort Pierce virus (accession: NC_025381), HLSV: Hibiscus latent Singapore virus (accession: NC_008310), CLIYMV: Clitoria yellow mottle virus (accession: NC_016519), SHMV: sunn-hemp mosaic virus (Accession: GCA_002866985), MMV: maracuja mosaic virus (NC_008716), PaFMV: passion fruit mosaic virus (NC_015552).

**Figure 6 plants-11-02166-f006:**
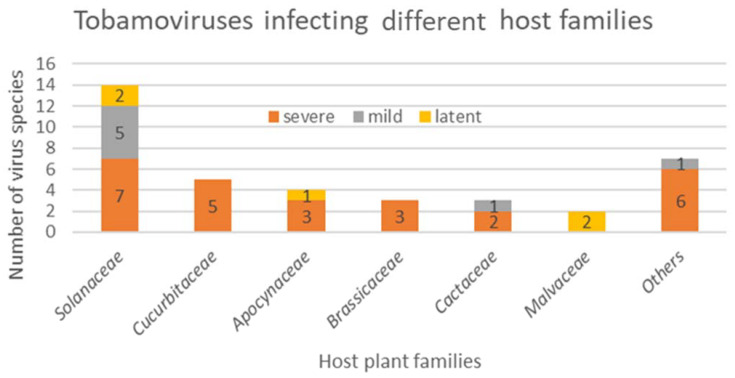
Overview of severe, mild and latent infections in different plant families caused by tobamoviruses mentioned in Figure 5. Severe, mild and latent infections have been defined based on the symptoms caused by the original host plant for each virus. It highlights the abundance of virus species and knowledge related to solanaceous-infecting tobamovirus and lack of studies on ornamental and wild host plants.

**Table 1 plants-11-02166-t001:** Division of tobamoviruses into subgroups based on genome organization and host range.

Subgroup	Location of OA	Overlapping ORFs	Host Range
I	Within ORF encoding movement protein (MP)	No	*Solanaceae* and *Orchidaceae*
II	Within ORF encoding coat protein (CP)	No	*Cucurbitacea* and *Fabaceae*
III	Within ORF encoding MP	Overlap of 77 nt between ORFs encoding MP and CP	*Cruciferaceae* and *Plantaginaceae*

## Data Availability

Not applicable.

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
