# Peer review of "To Be Seen or Not to Be Seen: Latent Infection by Tobamoviruses"

_plants, 2022, doi:10.3390/plants11162166_

Round 1

Reviewer 1 Report

The manuscript review the fact the latent infection of tobamoviruses. This kind of report is significant to readers and should be published in the Journal  MDPI.

ORV should be corrected as ORSV.

Author Response

ORV should be corrected as ORSV.

Corrected in figure and in the description. 

Reviewer 2 Report

This review focuses on the interaction of tobamoviruses with their plant hosts with the emphasis on the latent infections. This constitutes an interesting and poorly documented subject and a good basis for a review.

However, there are some parts in the manuscript which need attention (the lines in the manuscript are not numbered, but I hope you will be able to localise them)

Abstract

- “more and more viruses are being found which were previously hidden because of absence of obvious viral symptoms” this sentence is only partially correct and need rewriting. In fact, in many cases, HTS enabled to reveal complex viral infection, without possible link of individual viruses to the disease etiology. Moreover, many wild plants and plants in the agroecological interface (with and without symptoms ) are analysed by HTS, leading to the virus discoveries. Therefore, it is not only “absence of symptoms” behind of increasing number of novel viruses

- “Different mechanisms behind latency will be discussed” unfortunately, no molecular mechanisms leading to the latency are discussed in the review (either from virus side, or from the plant side)

Page 1, chapter 1.1

- a reference is needed after the 3rd sentence to support the statements on the symptomatology

- complement the sentence “Tobacco mosaic virus (TMV) and tomato mosaic virus (ToMV) are the most extensively studied species and they commonly cause chlorosis, mosaic patterns and leaf distortion ON SUSCEPTIBLE HOSTS”

Page 2: reference Castello et al. should be numbered

Page 3, reference Grdzelishvili et al. 2000 (legend of Fig3) and Lewandowski 2008, Ishibashi and Ishikawa 2016) (bottom of the page 3) should be numbered

Page 4: chapter 1.4. should be logically exchanged with 1.3, as it seems more logical to speak about the molecular groups immediately after genome and molecular properties

Page 5: Legend of Figure 5. Phylogenetic analysis of “latent and other tobamoviruses” … This is somewhat confusing, I suggest using  simply “Phylogenetic analysis of TOBAMOVIRUSES”, as no additional information is given, which of them are “latent” or “other” tobamoviruses

Page 6, Table 2: what does it mean “The virus remains in a dormant state”? it sounds strange

Table 2:

I consider Table 2 an essential element of this review, unfortunately, the whole Table is based solely on 2 references and in my opinion, it requires much more attention:

- e.g. concerning the “Viral persistence”, many persistent (cryptic) viruses replicate in high titres

- also, what about viruses with the integrated genomes?

- what about the hypersensitivity (is the same as resistance in your Table)?

Can authors provide, together with 2 very general references 32/33, an additional new column providing examples and specific references related to tobamoviruses for each case of “terminology”?

Page 6, last line: detail, to which virus the work of Malpica et al. refers

Page 7: Genetic variation: the problem of attenuated or mild strains/isolates within a virus species should be discussed more deeply. In fact, it is difficult to categorise the virus “damaging” and “latent” as the same virus can be symptomatic and asymptomatic on the same host depending on the concrete virus isolate/variant or a various cultivars of the same plant. Even within TMV or ToMV the mild isolates exists on the same host. From point of view of the authors, I feel that their perception of a “damaging” virus species is those whose at least some viral variants cause symptomatic reaction on a host (is my understanding correct?)

There is another (not discussed) point. A virus isolate can be asymptomatic in one host, but symptomatic in another host. Will we consider this isolate as symptomatic or asymptomatic?

Last point to be discussed in chapter 1.6 is the role of “helper viruses” in attenuation or increased symptomatology, specifically focusing on tobamoviruses

Page 8: “case study Ilyas 2021” is a reference?

Figure 6 (page 9) is obscure. How the “severe, mild, and latent infection” is defined? Based on which criteria? Instead of this graph, a Table would be more interesting for a reader, however, with concrete viruses listed (and explanation concerning such labelling)

As a general comment, I recommend to check the use of the term “strain/isolate/variant” throughout the manuscript as sometimes misleading.

Chapters 1.10.-1.12 should be 1.9.1 – 1.9.3

Round 2

Reviewer 2 Report

All my suggestions have been implemented in the new version of manuscript.